# Operational Excellence within Sustainable Development Concept-Systematic Literature Review

**Daniel Wojtkowiak ***  **and Piotr Cyplik**

Faculty of Engineering Management, Poznan University of Technology, 60-965 Poznan, Poland;
piotr.cyplik@put.poznan.pl
**\*** Correspondence: daniel.s.wojtkowiak@doctorate.put.poznan.pl; Tel.: +48-519-328-678

**Abstract:** The following paper was developed with a view to identifying the relationship between Sustainable Development and Operational Excellence concepts and to assess the state-of-the-art in Operational Excellence within Sustainable Development concept. Highly unstable business environments and opportunities occurring in the market require a specific approach and knowledge to achieve success. It is the authors' view that improvements in the delivery of sustainable results through social, environmental, and economic responsibility is possible thanks to modern management concepts, strategies, and tools. These activities and approaches, when properly incorporated, significantly support the functioning of businesses in a responsible and profitable way. This paper outlines a systematic literature review of the subject described above. The literature review has been conducted with the use of two databases: Web of Science and Scopus. The results of the analysis show that there is a visible connection between Operational Excellence and Sustainable Development concepts. On the basis of the review, one can notice a strong relationship between Operational Excellence and management concepts, e.g., Lean Management. Nevertheless, the input of the selected articles based on abstract analysis compared to the whole compilation concerning Sustainable Development shows that there is a gap. The authors believe that this topic is interesting and worth further analysis.

**Keywords:** sustainable development; sustainable manufacturing; operational excellence; systematic literature review; supply chain management

## 1. Introduction

In the era of global economic growth and parallel increase in labor abundance with less energy and material abundance, change in managing and thinking about industrial systems seems to be advisable [1]. Growing awareness and concern about the waste of resources, environmental degradation, and social issues are leading to the transition to more sustainable systems [2]. A growing need to integrate environmentally, socially, and economically sound choices into the business environment and supply chains is noticeable [3]. In view of the above, the authors suggest that ineffective and inadequate management may constitute the source of current challenges. Additional requirements and responsibilities that have emerged in recent years rest on those who have the greatest impact on the spheres of life in question. The economic paradigm as the key to making business decisions was discussed and analyzed by John Elkington in 1997. In his work, Elkington tried to find a solution and present a concept metaphorically known as triple bottom line where companies implement a decision-making approach that stands for equal significance in economic, environmental and social impact of their activity [4,5]. Taking into consideration that the current global challenges discussed above are the effects of the approaches that have been used in business activity, proper strategy, and different thinking about industrial systems should be implemented on a wider scale. According

to the authors, the changes in management approach implemented into manufacturing systems to increase sustainability do not have to be radical and economically unprofitable.

The authors suggest that the concept of Operational Excellence, which focuses on the effectiveness of business processes and efficiency increase, is in line with Sustainable Development mainly because of waste decrease and better use of resources. As an example of this, a positive relation between lean management and sustainable manufacturing systems has been discovered. The adoption of the Lean Management concept supports sustainability practices [6,7]. Moving forward, the authors notice links between more effective business processes resulting from Operational Excellence and the economic, environmental, and social objectives of Sustainable Development.

The roots of the Operational Excellence term can be found in the 18th century work of Adam Smith entitled "An Inquiry into the Nature and Causes of the Wealth of Nations" which is considered to be the foundation of classical economics and covers such topics as free market, productivity, and division of labor. It is treated as the first scientific analysis of the economic phenomena visible in societies. In further years, industrialists such as Henry Ford with his first vehicle assembly line and Frederick Winslow Taylor with his publication "The Principles of Scientific Management" both significantly influenced the creation and development of such fields as industrial engineering, focusing on optimizing complex processes and systems by developing, improving, and implementing integrated systems of people, money, knowledge, information, equipment, energy and materials which lead to Operational Excellence [8,9]. The term 'Operational Excellence' can be defined in various ways. According to Naftanaila and Cioana, Operational Excellence is a crucial factor in achieving business success, as it is related to the company's operation with customers. It is a philosophy at the lowest operating level which is closest to resources directly engaged in business operations, but it is critically connected with the company's strategy level [10]. Conversely, Croom et al. show that there are three main areas of focus captured in the dimension of Operational Excellence: right first time, high productivity, and effectiveness (customer/market oriented) of processes [9]. The term Operational Excellence is also understood as a state that includes the use of industrial activities and concepts like Lean, Six Sigma, Reverse Logistics, the Internet of Things, Industry 4.0, Information Technologies and Business Process Reengineering as a means or way of obtaining proper economic results [11]. Improvement of the organizations' competitiveness is seen as a result of appropriate decisions made by the organizations and is a reflection of operational efficiency [12,13]. Improvement and progress as a product of Operational Excellence is also very often associated with innovation, new technologies, and solutions. Innovativeness usually allows lower business operating costs and increased net profit [14]. Thus, achieving Operational Excellence can be associated with the implementation of individual management concepts and tools that determine an efficient system that allows an appropriate cost level to be obtained and an innovation increase that results from investments and continuous improvement process through problem solving. In this paper, the authors understand Operational Excellence as a means of achieving the organization's objectives. It focuses primarily on increasing the efficiency of the processes and achieving optimal results in the organization. At the same time, optimal usage of the available resources and loss elimination is noticed [15,16]. The literature shows that achieving Operational Excellence is closely connected with the social aspects of the organizations. According to Theodore J. and Anderson C., there are four crucial factors that contribute to achieving Operational Excellence. The first one is connected with the organization's vision which should present the desired state. The second point includes the engagement of people in strategy execution. The third one demonstrates that the performance of the processes should fulfill the stakeholders' requirements. The last point is about technology which should allow people to achieve set objectives efficiently [17].

Sustainable Development as an idea has been the subject of scholarly research since the late 1960s. Its main focus has been decreasing global pollution [1]. However, the term not only concerns the area of industry and its environmental influence, but is a much broader concept concerning social and economic aspects of human activity and life. Sustainable Development can be described as a system designed to satisfy the growing needs of the global population with the simultaneous preservation of

the environment [18–20]. Gro Harlem Brundtland defines Sustainable Development as "development that meets the needs of the present without compromising the ability of future generations to meet their own needs" [21]. Another commonly used definition comes from Paul Hawken and presents the basic assumptions of the concept in a very simple and accessible way; "Leave the world better than you found it, take no more than you need, try not to harm life or the environment, make amends if you do" [22]. Taking into consideration the scope and width of access, Sustainable Development has become a buzzword in the global discourse concerning proper directions of development [23].

Thailand, and other countries where economic development is directly connected with energy demand and usage, can serve as an example [24]. Consider energy, whose usage in most cases increases carbon emissions, pollution, and other greenhouse gas emissions, in which case converting it into Sustainable Development practices should be taken into account [25]. It is specifically instinctive when industries strive to integrate the three main pillars (i.e., social, environmental, and economic aspects) of sustainability into its strategies [3]. Green, sustainable solutions, e.g., in the field of logistics, not only positively affect the environment due to reduced emissions of carbon dioxide and greenhouse gases but also improve economic performance and help achieve a stronger market position [26]. Taking into consideration the fact that the industry sector has, most likely, the biggest influence on the three main areas of Sustainable Development (SD), i.e., society, environment and economy, through the years, additional sub-concepts and terms related to Sustainable Development have been created. The example is Sustainable Manufacturing which directly refers to SD, and the United States Department of Commerce defines it as a "manufacturing processes that minimize negative environmental impacts, conserve energy and natural resources, are safe for employees, communities and consumers and are economically sound" [27]. Another example is Sustainable Supply Chain Management or Green Supply Chain Management which refers to "integrating environmental thinking into supply chain management, including product design, material sourcing and selection, manufacturing processes, delivery of the final products to the consumers, and end-of-life management of the product after its useful life" [28,29]. Another definition found in the literature says that GSCM (Green Supply Chain Management) is "The practice of monitoring and improving environmental performance in the supply chain . . . " [30].

The United Nations set 17 Sustainable Development Goals to reach by 2030. These goals are composed of 169 targets and 232 indicators [31]. All of the goals refer directly to the three main pillars described above in this paper. The structure of the Sustainable Development Goals dashboard is: Goal/Target/Indicator. To present an example, one of the goals have been described as follows:

- Goal 7—Ensure access to affordable, reliable, sustainable and modern energy for all.
- Target 7.3—By 2030, double the global rate of improvement in energy efficiency.
- Indicator 7.3.1—Energy intensity measured in terms of primary energy and GDP [32].

This breakdown shows that Sustainable Development can be measured and influenced through activities in countries, markets, industries or by specific manufacturers. To cascade it on the operational levels, more precise performance indicators and targets need to be set locally.

The authors understand Sustainable Development as a much wider concept than Operational Excellence but see some links and their complementarities. According to the information above, the authors compare a concept that has its roots in the 18th century and focuses mainly on economic efficiency in industry to the one which is much broader in its influence but far less mature and is still being gradually developed and understood. For the purpose of showing complementarity, this paper seeks to analyze current state-of-the-art through the method of systematic literature review.

## 2. Materials and Methods

Having noticed the potential correlation between the two different concepts, the authors posed the following research questions:

1. Is there a positive correlation between Operational Excellence and Sustainable Development?

2.    What specific element of Operational Excellence correlates with Sustainable Development the most?

To answer these questions a systematic literature review and literature analysis method has been chosen.

A systematic literature review is one of the methods to analyze and provide a theoretical background for the research. According to Fink's definition, a literature review should be a "systematic, explicit, and reproducible method for identifying, evaluating, and synthesizing the existing body of completed and recorded work produced by researchers, scholars, and practitioners." [33]. Petticrew and Robert state that systematic literature review is essentially a tool. The question they ask is whether the selected tool is always right for a given job [34]. It can be concluded that systematic literature review is not always the right choice. Nevertheless, the process of systematic review has played a major role in evidence-based practices for at least two decades [35]. Undertaking systematic review is now regarded as a "fundamental scientific activity" [36]. It also allows to learn the scope of the research, show and understand what existing literature and research say about the specific matter. It allows one to identify, select and critically evaluate literature in the chosen research areas in a rigorous and systematic manner [37]. It allows the review of the literature in a specific way and gives the possibility of answering the set questions and hypotheses and decide if they are correct or not. The form of the systematic literature review facilitates filling the gap in methods designed for analyzing problems in the specific research or literature area [1,35].

The research made by authors covered the search of the phrase "Operational Excellence" in two databases: Scopus and Web of Science. With a view to making the research and analysis more precise, subject areas have been set.

Subject area (Scopus): Engineering; Business Management and Accounting; Energy; Computer Science; Chemical Engineering; Earth and Planetary Sciences; Decision Sciences; Social Sciences; Materials Science; Economics, Econometrics and Finance, Environmental Science; Chemistry; Physics and Astronomy; Biochemistry, Genetics and Molecular Biology; Agricultural and Biological Sciences; Multidisciplinary.

The subject area includes (Web of Science): Management, Environmental Studies, Operations Research Management Science, Engineering Industrial, Statistics Probability, Business, Transportation Science Technology, Engineering Manufacturing, Water Resources, Engineering Electrical, Electronic, Computer Science Information Systems, Computer Science, Interdisciplinary applications, Energy Fuels, Computer Science Cybernetics, Engineering Multidisciplinary, Transportation, Engineering Aerospace, Computer Science Artificial Intelligence, Engineering Marine, Computer Science Theory Methods, Engineering Mechanical, Engineering Chemical, Engineering Civil, Economics, Materials Science Characterization Testing, Environmental Sciences, Multidisciplinary Sciences, Information Science Library Science, Automation Control Systems, Computer Science Software Engineering, Agricultural Economics Policy, Green Sustainable Science Technology, Business Finance, Engineering Environmental, Physics Applied, Engineering Petroleum, Chemistry Analytical, Materials Science, Multidisciplinary, Metallurgy, Metallurgical Engineering, Materials Science Ceramics, Telecommunications, Materials Science Composites, Social Sciences Interdisciplinary, Materials Science Textiles, Biotechnology Applied Microbiology, Chemistry Multidisciplinary, Construction Building Technology.

Despite the fact that subject areas are not identical in both databases, the authors selected subjects that cover the same or similar fields.

In the next step of the analysis, extracted articles from both databases have been processed through the steps of the authors' algorithm and prepared for abstract analysis. The authors set the keywords' criteria for additional deeper selection. To achieve the results, the authors analyzed it according to their interest field, so the abstracts have been analyzed emphasizing the search for a connection with such subjects as: Sustainability and Environment, Lean and Agile Management, Maturity, ISO Standards, Big Data, Business Intelligence and Automation.

In order to present the logic and the subsequent steps of the analysis, the graphical representation prepared as an algorithm is shown below in Figure 1.

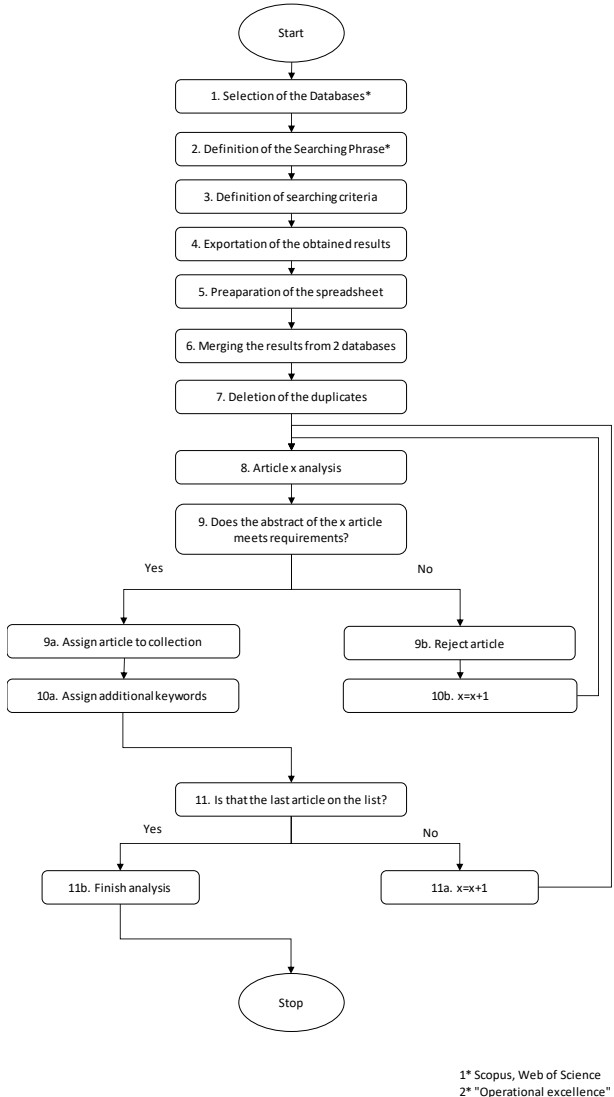

**Figure 1.** Systematic literature review logic algorithm. Source: Own study based on Odważny, F., Wojtkowiak, D., Cyplik, P., Adamczak, M., 2018, p. 470.

## 3. Results

### 3.1. Preparation for the Analysis

In the first search for the phrase "Operational Excellence" without any limitations, the database produced 1916 results. The Scopus database searched titles, abstracts, and keywords that contain either "operational" or "excellence" and treated these words as separate phrases which means that the database showed the results containing at least one of the two phrases. After the introduction of the first search parameters, i.e., searching the exact phrase "Operational Excellence", Scopus produced 1332 results. The next parameter set was English language only and subject area as presented in Chapter 2. As a result, 1210 publications meeting the above-described requirements were produced.

As with the Scopus database, the Web of Science search for "Operational Excellence" treated as two separate phrases resulted in 815 publications. The first limitation which was treating "Operational Excellence" as one phrase produced 422 publications. The next criterion focused on English language

only papers and subject area as described in Chapter 2, which gave the final result of 357 publications that met the requirements.

The combination of the extracted results from both sources produced 1567 articles which were analyzed in the context of subject compliance. The next step was finding and erasing duplicating articles from the list. The final list of unique articles which were focusing on "Operational Excellence" stood at number of 1269. Figure 2 shows the subsequent steps and results of the preparation.

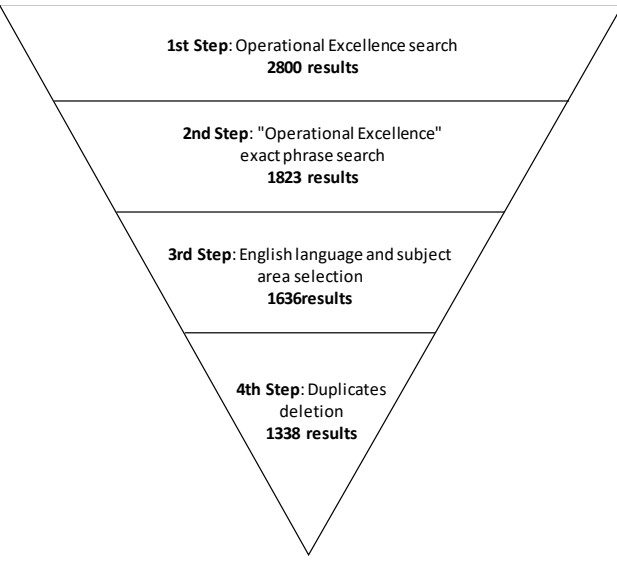

**Figure 2.** Preparation for the analysis steps. Source: Own study.

*3.2. Results of the Analysis*

3.2.1. Analysis of the Article Collection

Analyzing the number of articles about Operational Excellence in the compiled collection shows that the term has been used in science since 1987. The chart showed in Figure 3 suggests that the concept began garnering interest in 2007 when the number of publications had doubled in comparison to previous years, gradually growing until 2020. Since 2016 the average number of publications in this field exceeded 100 articles per year. In August 2020 there were already 80 articles about Operational Excellence which suggests yet another increase in interest in this area.

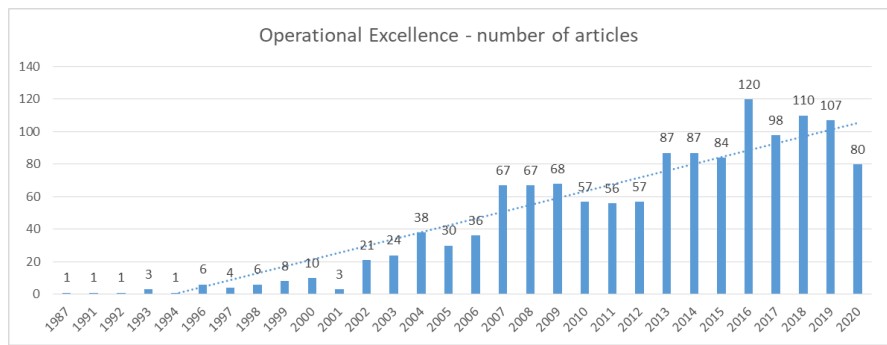

**Figure 3.** Operational Excellence–the number of articles over the years. Source: Own study.

The analysis conducted by the authors covered the connection between Operational Excellence and Sustainability with Environment, but also; Lean, Agile, Maturity, ISO Standards, Big Data, Business Intelligence and Automation.

The keywords have been selected according to the authors' interests and earlier publications as described in Chapter 2. Figure 4 shows that the keyword most often connected to the Operational Excellence is the concept of Lean which seems to be understandable. When briefly describing the concept of Lean, it can be stated that its assumptions refer to the continuous improvement and achievement of process excellence by reducing losses in the processes. The next most frequently linked keywords included: Sustainability and Environment which the authors decided to treat as one. The term Operational Excellence is also closely connected with the automation of the processes. Looking at the Table 1 below which shows number of articles published through the years, it can be seen that the significant rise of selected keyword correlation to Operational Excellence have been seen since 2008 and have gradually grown through the years. The most common connection to Operational Excellence in last 2 years are: Lean Management, Sustainability and Environment. Based on this it can be concluded that there are some significant interconnections between these areas and that these elements influence on each other.

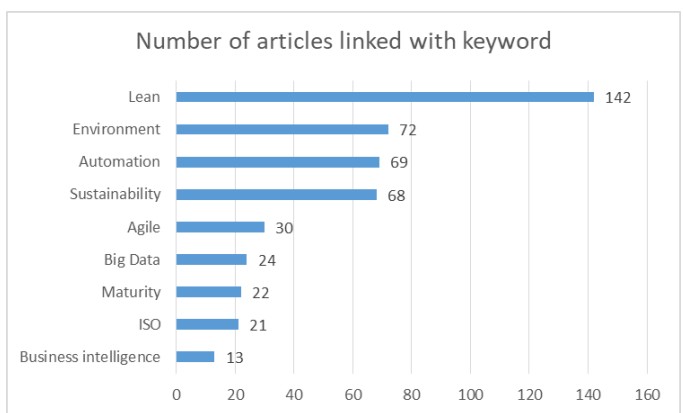

**Figure 4.** The number of Operational Excellence articles linked with selected keywords. Source: Own study.

**Table 1.** Selected keywords in Operational Excellence articles through years.

| Year | Lean | Environment | Automation | Sustainability | Agile | Big Data | Maturity | ISO | Business Intelligence | Total |
|---|---|---|---|---|---|---|---|---|---|---|
| 1992 | 0 | 1 | 0 | 0 | 0 | 0 | 0 | 0 | 0 | 1 |
| 1996 | 0 | 1 | 0 | 1 | 0 | 0 | 0 | 0 | 0 | 2 |
| 1997 | 0 | 0 | 0 | 0 | 0 | 0 | 0 | 0 | 0 | 0 |
| 2000 | 0 | 0 | 0 | 0 | 1 | 0 | 0 | 1 | 0 | 2 |
| 2001 | 1 | 0 | 0 | 0 | 0 | 0 | 0 | 0 | 0 | 1 |
| 2002 | 0 | 0 | 1 | 0 | 0 | 0 | 0 | 1 | 0 | 2 |
| 2003 | 1 | 0 | 4 | 0 | 0 | 0 | 0 | 0 | 0 | 5 |
| 2004 | 3 | 4 | 2 | 0 | 0 | 0 | 0 | 0 | 1 | 10 |
| 2005 | 2 | 1 | 3 | 0 | 0 | 0 | 1 | 1 | 0 | 8 |
| 2006 | 0 | 2 | 2 | 1 | 0 | 0 | 0 | 1 | 0 | 6 |
| 2007 | 4 | 2 | 4 | 1 | 0 | 0 | 1 | 1 | 0 | 13 |
| 2008 | 8 | 5 | 4 | 1 | 0 | 0 | 0 | 1 | 0 | 19 |
| 2009 | 7 | 4 | 4 | 4 | 0 | 0 | 0 | 0 | 2 | 21 |
| 2010 | 7 | 4 | 6 | 5 | 1 | 0 | 1 | 0 | 0 | 24 |
| 2011 | 5 | 1 | 2 | 1 | 1 | 0 | 1 | 0 | 0 | 11 |
| 2012 | 5 | 3 | 2 | 1 | 4 | 1 | 0 | 0 | 2 | 18 |
| 2013 | 6 | 5 | 5 | 3 | 1 | 0 | 0 | 0 | 3 | 23 |
| 2014 | 9 | 8 | 4 | 7 | 1 | 0 | 1 | 1 | 1 | 32 |
| 2015 | 8 | 5 | 4 | 5 | 0 | 0 | 4 | 0 | 1 | 27 |
| 2016 | 11 | 4 | 6 | 5 | 4 | 1 | 5 | 3 | 2 | 41 |
| 2017 | 8 | 9 | 1 | 5 | 5 | 3 | 1 | 3 | 0 | 35 |
| 2018 | 16 | 2 | 5 | 5 | 2 | 7 | 2 | 2 | 1 | 42 |
| 2019 | 12 | 7 | 6 | 9 | 3 | 2 | 1 | 0 | 0 | 40 |
| 2020 | 29 | 4 | 4 | 14 | 7 | 10 | 4 | 6 | 0 | 78 |
| Grand Total | 142 | 72 | 69 | 68 | 30 | 24 | 22 | 21 | 13 | 461 |

The colors help to show the increase the number of articles that contain selected keyword related to Operational Excellence through years.

### 3.2.2. Sustainability within Operational Excellence

In the next step of the systematic literature review analysis, the authors focused on articles linked with sustainability and environment as terms related to the concept of Sustainable Development. From both databases, among articles which discussed Operational Excellence and were connected with Sustainable Development, the authors extracted 128 positions which is 10% of the prepared collection of 1269 articles and 4.69% of the primary search for Operational Excellence. To visualize the steps used in this paper, the authors prepared a graphic representation of the results through the following steps shown in Figure 5 below. The graphic is the extended version of the one presented in Chapter 3 and contains an additional fifth step focusing on Sustainability and Environment.

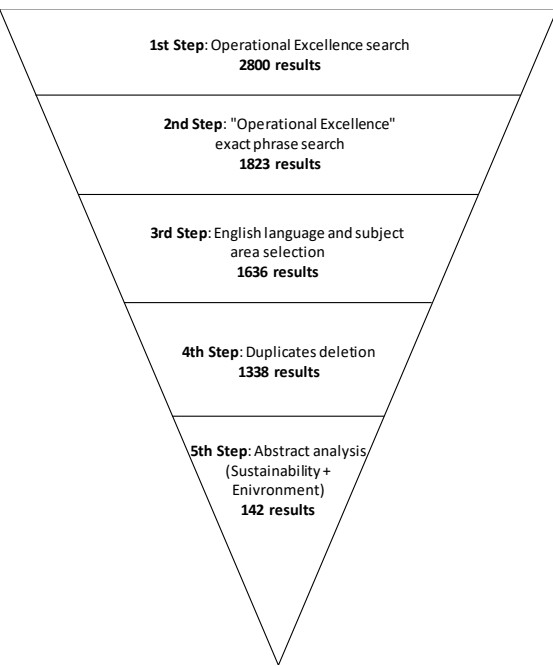

**Figure 5.** Analysis results–Steps. Source: Own study.

For a better understanding of the connections between Sustainable Development and Operational Excellence, the authors analyzed the interest of the topic over the years. As Figure 6 shows since 2014, a rise in the number of publications can be seen. After 5 years, 2019 brought the highest number of publications. In August 2020, 20 articles were already published which translated into almost 125% of whole 2019 collection. That means the rising trend in this matter is continuing and is predicted to be strengthened.

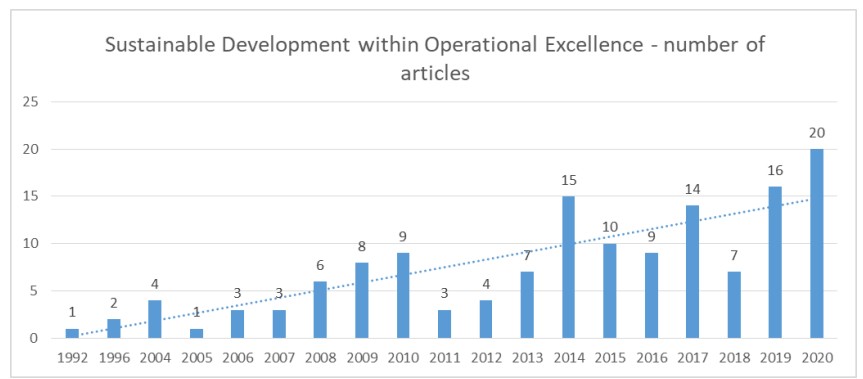

**Figure 6.** Sustainable Development within Operational Excellence-Number of articles chart. Source: Own study.

Another study level focuses on the origin of the published articles and shows that over one third of the analyzed articles come from the United States of America and the United Kingdom as shown in Figure 7.

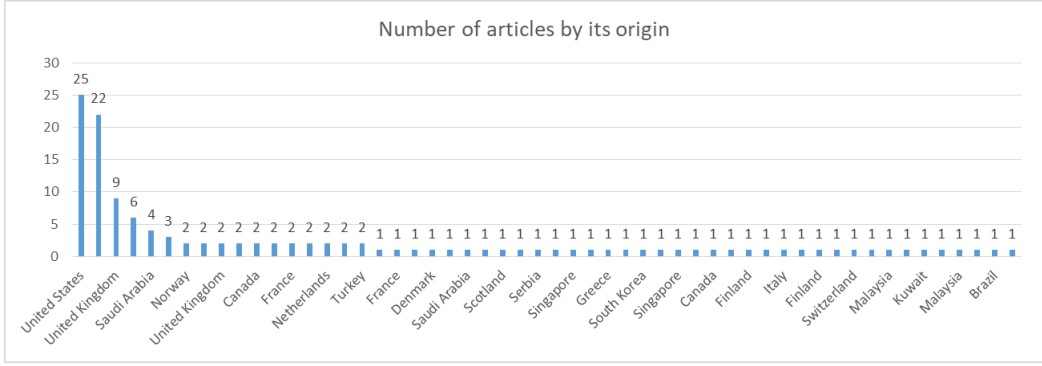

**Figure 7.** Number of chosen articles by origin. Source: Own study.

What is more, the same phenomenon of strong United States participation can be seen in the whole analyzed collection, where the number of articles represents almost 30%.

In the last part of the analysis, the authors checked the connection of the analyzed 142 articles concerning Sustainability within Operational Excellence with the selected keywords. Figure 8 shows that the most common connection to the area is Lean Management which suggests that this management concept is most commonly used in terms of Operational Excellence. As shown in the introduction chapter, a positive relation between Lean and Sustainable Manufacturing has been proven [4].

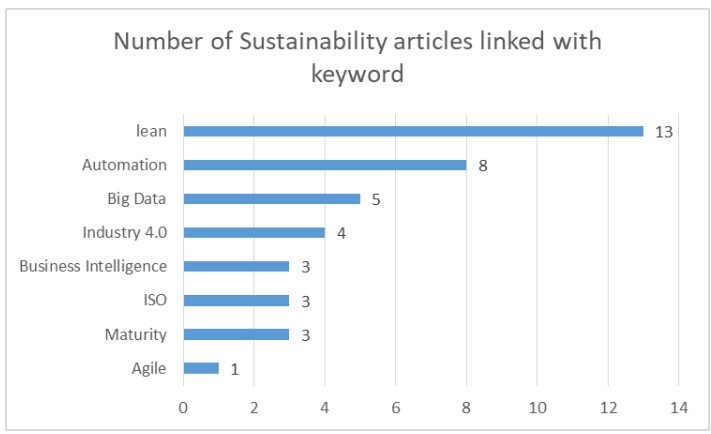

**Figure 8.** Operational Excellence within Sustainability articles linked with keywords. Source: Own study.

After abstract analysis of the selected 13 articles which refer to Lean Management, the authors underline the rising significance of Sustainability and suggest implementing Lean practices.

## 4. Discussion

The results obtained in this paper show that there is a visible connection between Operational Excellence and Sustainable Development which can be proven by the selected 142 articles. It means that 1 out of 10 articles about Operational Excellence refers to Sustainable Development. What is also worth highlighting is the fact that 10% of the selected 142 articles show a direct connection between Lean Management and Sustainable Development, showing Lean concept as a way of increasing sustainability in a company [38–40]. On the other hand, looking at the total number of articles concerning Sustainable Development found in Scopus and Web of Science databases in the number of

nearly 200,000, the selected 142 articles represent only 0.07% of the whole collection which can suggest a gap to consider.

The results are important because they show that achieving or increasing sustainability does not have to be connected with drastic changes and high costs. It can be achieved step by step through very well-known management techniques that can ease the process of introducing green policy [41,42]. The authors are aware that it is not possible to achieve full sustainability only by implementing management techniques and the results do not show great significance [43]. Investments in new sustainable technical solutions is required in the long term. Additionally, the process of becoming sustainable can be profitable not only for the environment but also for industries that would use the required resources, optimizing cost through a decrease in process losses in a wiser and more effective way [44,45].

The analysis does not show what specific tools or techniques should be used to increase sustainability. It shows that the idea of system process efficiency improvement and waste decrease is in line with the concept of Sustainable Development. Specific tools, management techniques, IT solutions and analysis required for improving processes depend on specific organizational units like manufacturing plants.

The authors believe that the concept of Sustainable Development can be strengthened through the promotion in manufacturing areas of simple rules referring to everyday habits which are understandable for all employees. Another strengthening factor is a closer relationship with suppliers [46]. Combined together, they will allow an increase in the awareness of sustainability and accelerate the mindset change which seems to be the most important aspect in the process.

## 5. Conclusions

The main objective of this paper was to present the linkage and complementarities of two different concepts: Sustainable Development and Operational Excellence, and answer two research questions:

1. Is there a positive correlation between Operational Excellence and Sustainable Development?
2. What specific element of Operational Excellence correlates with Sustainable Development most?

A systematic literature review and an analysis have been conducted. This method helps to understand how specific keywords correlate with each other and allows us to define the research gaps.

To answer the first research question, the authors conducted the analysis which indicated some interesting positive correlation between two different concepts with different characteristics and applications. The link between the concepts has been shown and proven. Over 10% of the analyzed article collection about Operational Excellence has a direct connection with Sustainable Development. The answer to the second research question was possible to obtain through keyword and abstracts analysis. Lean Management concept, which can be considered as one of the elements of Operational Excellence, most often correlates with the analyzed field. The authors see the potential in driving Sustainable Development objectives through well-known practices and tools used in Lean Management. Sustainable Development is a much wider term than Operational Excellence which should be understood as a tool or one of the ways of achieving Sustainability. Taking into consideration the growing consciousness of customers around the world and current knowledge about the condition of the environmental pollution [47], it became important for companies to think about the environment [48]. The authors believe that strategies of the organizations in industry business should be set in line with the Sustainable Development concept due to the benefits that it brings for both the environment and companies. Using resources like energy, which is a vital resource for economic growth, is positively correlated with business operations and negatively impacts the environment [24], which suggests that being operationally excellent with resource usage is economically arguable and gives multiple benefits for both the organization and the environment [49]. Organizational maturity measurement should help in designing and incorporating proper business strategy that allows one to reach the objectives [50].

Management concepts, tools, and technology should be used to operate and work in a certain way to achieve the Operational Excellence as well as Sustainable Development Goals [51].

This article shows there is a visible connection between these two concepts which can be driven and implemented in parallel. The main contribution of this article to existing literature is the summary of current state-of-the-art and global trends in achieving sustainability through the use of technological development and management techniques.

The authors believe that there is a need for further development and deeper analysis of the field and mutual influence of these two concepts on each other. The paper shows there is a potential for deeper analysis of the SD (Sustainable Development) absorption into company through Operational Excellence implementation.

**Author Contributions:** Conceptualization, D.W. and P.C.; methodology, D.W. and P.C.; software, D.W.; validation, D.W.; formal analysis, D.W.; investigation, N/A; resources, D.W. and P.C.; data curation, D.W.; writing—original draft preparation, D.W.; writing—review and editing, D.W. and P.C.; visualization, D.W.; supervision, P.C. All authors have read and agreed to the published version of the manuscript.

**Funding:** This article was funded by Poznan University of Technology, Faculty of Engineering Management [project number: 0812/SBAD/4178.]

**Conflicts of Interest:** The authors declare no conflict of interest.

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
