# Peer review of "Operational Excellence within Sustainable Development Concept-Systematic Literature Review"

_sustainability, doi:10.3390/su12197933_

Round 1
Reviewer 1 Report
This paper examine relationship between sustainable development and operational excellence concepts and to assess the state of the art in operational excellence within sustainable development concept using systematic literature review.
To be accepted at this journal following issues have to be addressed.
To be accepted at this journal following issues have to be addressed.
- Abbreviation should be spelled out. For example, SLR at line 105.
- Author should argue motivation and implication of this study. –
- I cannot find contribution of this study.
- This paper only search database with some related keywords, such as operational excellence and sustainability.
- Should do some analytic test. For example, at figure 3, 7, is the slope of line significant?
- Are they increase over the year?
- Also, authors could address implication from results at conclusion. What is implication of linked keywords?
Author Response
Dear Reviewer,
Thank You for your contribution and comments given to my work.
I used it for the review and corrections.
I am sending improved article. Hope it You'll find better and more suitable for publication.
Best regards,
Daniel Wojtkowiak

Reviewer 2 Report
Dear authors, your paper analyses an interesting field. However at the current stage, it is a very preliminary research.
In particular, the analysis is based on a merely statistical analysis. In addition, the discussion is very weak. On the point, a good literature review requires a deep analysis of the main findings and future research directions.
Author Response
Dear Reviewer,
Thank You for your contribution and comments given to my work.
I used it for the review and corrections.
I am sending improved article. Hope You'll find it better and more suitable for publication.
Best regards,
Daniel Wojtkowiak

Reviewer 3 Report
Dear authors!
The study aim is to attempt identification of the relationship between Sustainable Development and Operational Excellence concepts and to assess the state-of-the-art in Operational Excellence within Sustainable Development.
This issue is rather interesting for readers, at least for the group of researchers who are dealing with these topics. The research are not presented in a quite clear way and the authors should better explain at the beginning what exactly connection is between Sustainable Development and Operational Excellence concepts and why they decided to examine exactly this relationship?
I presented my other specific comments below.
Abstract: Mostly this part of the text is presented as a whole, so I guess the authors should consider to cut this part of the text a little and present this information as one coherent text.
Introduction: The sentences: “In the era of worldwide economy growth, use of the Operational Excellence implementation approach described above, Sustainable Development concept execution within manufacturing companies can be considered.” and “Authors understand Sustainable Development as much wider term than Operational Excellence but see some links and complementarities in both” – do not explain well the links and complementarities between these two concepts. Thus the key links and complementarities should be listed in this part of the text and explained wider.
The chapter titled: 2. Materials and Methods is well structured and prepared according to the chosen methodology key assumptions.
The last part: Discussion should be extended and the conclusions as well as the further research directions should be clearly and separately presented. The authors should consider to separate the conclusions as the final chapter of the paper.
Taking into account the paper topic (strongly theoretical) the theoretical background should be enriched and the bibliography should be extended much.
Generally the issue is interesting but needs to be corrected at least according to the above remarks.
Good luck!
Author Response

(The authors gave the same response as above.)

Round 2
Reviewer 2 Report
well done
Author Response
Good evening,
I am sending revised paper with language corrections.
Regards,
Daniel Wojtkowiak
Reviewer 3 Report
Dear Authors
In the updated texts still some references are missing (see the bibliography, some positions are empty). I think that the number of cited publication is still very low on comparison with other papers and the final part - the conclusions is not enough highlighted.
Author Response
Good evening,
I am sending revised paper with language corrections.
Language corrections have been made in cooperation with specialist.
Changes tracking is availlable within file.
Regards,
Daniel Wojtkowiak